# Optimising Mechanical Ventilation for Indoor Air Quality and Thermal Comfort in a Mediterranean School Building

**Krista Rizzo, Mark Camilleri, Damien Gatt and Charles Yousif \***

Institute for Sustainable Energy, University of Malta, MXK 1531 Marsaxlokk, Malta;
krista.rizzo@um.edu.mt (K.R.); mark.camilleri@rews.org.mt (M.C.); damien.gatt@um.edu.mt (D.G.)
\* Correspondence: charles.yousif@um.edu.mt; Tel.: +356-23407831

**Abstract:** The growing concern over indoor air quality (IAQ) and thermal comfort in classrooms, especially post-COVID-19, underscores the critical need for optimal ventilation systems to bolster students' health and academic performance. This study explores the potential for improving indoor air quality and thermal comfort in the most energy- and cost-optimal manner using a demand-controlled ventilation (DCV) system coupled with a carbon dioxide control sensor. This is achieved through precooling via night purging in summer and by introducing warmer corridor air into the classroom in winter. The methodology employs both computer simulation and a real-world case study. The findings reveal that while natural ventilation in winter can achieve IAQ standard (EN 16798-1) thresholds for classrooms under favourable outdoor conditions, it results in uncontrolled and excessive energy loss. The retrofitted DCV system, however, maintained $CO_2$ levels below the recommended thresholds for at least 76% of the year depending on classroom orientation and only exceeded 1000 ppm for a maximum of 6% of the year. This study also indicates that utilising the external corridor as a sunspace can further enhance the system's efficiency by preheating incoming air. This comprehensive study highlights the significant potential for integrating mechanical and passive solutions in school ventilation systems. This contributes to the attainment of the United Nations Sustainable Development Goal 11 and ensures healthier and more energy-efficient learning environments that benefit both students and the environment.

**Keywords:** indoor air quality; ventilation in schools; $CO_2$ emission reduction; thermal comfort





## 1. Introduction

Students spend a significant part of their day at school in classrooms. Poor indoor air quality (IAQ) and uncomfortable indoor temperatures in classrooms are serious global issues [1–3]. This is exacerbated when the ventilation rate in the classrooms is not sufficient to avoid overheating as well as to remove pollutants. This can happen for several reasons, for example, when windows are kept closed to avoid discomfort caused by external noise or to prevent draughts [1]. In some cases, the building characteristics do not allow for sufficient ventilation, such as when sliding windows are installed which only allow half of the window to be open at any one time. A case study in Malta showed that such a situation had resulted in higher indoor temperatures in the absence of climate control [4]. Moreover, the increasing incidence of respiratory symptoms being reported has brought greater attention to the importance of indoor air quality (IAQ) in schools, leading to a growing focus on research in this area [5]. Fsadni et al.'s [6] study showed that the school indoor environment has a direct impact on Maltese children's respiratory health, and recommendations were made to authorities to improve schools' IAQ.

Carbon dioxide ($CO_2$) concentration is a good indicator of indoor air quality and ventilation rates [7,8]. $CO_2$ concentration can be used as a tracer gas for human-related pollutants when building occupants are the main source of $CO_2$ generation [9]. Thus, $CO_2$ concentration is used as a control parameter to regulate the ventilation rate, known as

$CO_2$-based demand-controlled ventilation (DCV). Numerous studies, including those by Daisey et al. [10], have linked high levels of carbon dioxide ($CO_2$) in classrooms to negative effects on occupants, including health symptoms, decreased performance, and increased absence from school [10]. Elevated $CO_2$ concentrations exceeding 1000 ppm have been shown to cause various complaints such as headaches, nose and throat ailments, allergy, asthma, tiredness, lack of concentration, and fatigue [11–14]. Similarly, the investigation carried out by Bakó-Biró et al. [15] provided strong evidence that low ventilation rates in classrooms significantly reduce students' attention and vigilance and negatively affect memory and concentration. Krawczyk et al. [16] found a correlation between elevated $CO_2$ levels in classrooms and reduced satisfaction with indoor comfort among occupants. Coley et al. [17] demonstrated that high $CO_2$ levels negatively impact schoolchildren's cognitive function and found that when $CO_2$ levels increased, the attentional processes of students were significantly slower. This implies that high $CO_2$ levels in classrooms could potentially have detrimental effects on learning and educational attainment. Exposure to $CO_2$ levels above 1000 ppm increases the risk of developing dry cough and rhinitis [18] as well as increased transmission of infectious diseases including COVID-19 [19].

On the other hand, increasing the ventilation rate has been shown to enhance IAQ and thermal comfort and to positively impact various outcomes, such as reducing the spread of airborne infections [20], improving students' academic performance [21], and reducing the levels of indoor pollutants [22]. Schibuola and Tambani [23] deduced that having proper ventilation in classrooms lowers $CO_2$ levels and reduces chemical contaminant exposure, promoting good IAQ. Other studies indicate that increasing ventilation rates in classrooms leads to a 15% improvement in cognitive function scores among students [13], improved learning performance and pupil attendance [24].

Traditionally, schools in Malta have relied on natural ventilation by opening windows, which is common in temperate climates. However, natural ventilation may not always benefit IAQ, as outdoor air quality can be poor and occupants may prefer to keep fenestration closed to avoid heat loss during winter and maintain a more stable temperature, for noise reduction, or to avoid draughts during windy days. Studies have shown that internal $CO_2$ levels can be worse in countries where schools primarily rely on natural ventilation systems [25–27]. Schibuola and Tambani [23] found that natural ventilation via opening windows may not be sufficient to ensure appropriate IAQ. Their study found significant variability in $CO_2$ values in the same classroom caused by human behaviour, making natural ventilation difficult to predict and manage. One possible solution suggested the installation of a simple and cheap $CO_2$ meter in each classroom to inform teachers about ventilation needs. Additionally, while natural ventilation seems like the most energy-efficient way to achieve the required air changes per hour, it may result in heat loss during winter due to uncontrolled air changes, impacting indoor comfort and increasing energy consumption for heating [28].

Thermal comfort is another critical factor for a conducive learning environment [29]. Poor thermal conditions can negatively impact students' performance, leading to discomfort, fatigue, and reduced cognitive function [30,31]. Increased classroom temperatures and humidity, associated with ventilation, are linked to an increased incidence of allergic conditions in primary school children in Malta [4].

Climate change is expected to worsen thermal comfort in schools due to increased temperatures. An increase in the frequency and intensity of heatwaves and hot summers worldwide is anticipated [32,33], with some studies even suggesting that the increase will be higher than initially estimated [34]. According to a recently published report by the World Health Organization (WHO) [35], Malta's percentage of hot days (i.e., "heat stress") is projected to increase substantially from about 15% of all days on average between 1981 and 2010 to 80% by 2100 under a high-emissions scenario, which refers to a situation where greenhouse gas emissions are not effectively reduced. Even under a two-degree scenario, which assumes rapid emissions decrease, the WHO report projects a 40% increase in hot days by 2100. As students in classrooms are passive recipients of thermal conditions, with

limited opportunities for changing activity levels or adjusting environmental variables, this presents a significant challenge to their thermal comfort [36]. Therefore, in addition to addressing indoor air quality, improving thermal comfort in schools is also crucial for the well-being of students, particularly in Malta, where the effects of climate change are expected to be significant.

In Maltese schools, it is common practice to neither mechanically space-heat nor space-cool the classrooms except for the use of ventilation fans. In recent years and due to rising temperatures, there has been an increase in the number of schools being equipped with individual space-cooling and -heating air-conditioning systems. In 2019, a petition signed by nearly 5000 individuals called on the Maltese government to install space-cooling and -heating systems (the most common being split units or variable refrigerant flow (VRF) systems) in classrooms to alleviate the unbearable heat, particularly during the summer season [37]. While these systems have the potential to improve indoor thermal comfort with respect to indoor temperature, they can also result in high energy consumption. This occurs due to uncontrolled air changes and the resulting high heat gains or losses if the sole reliance is on natural ventilation through windows. Furthermore, poor IAQ will result if the windows are kept closed to prevent heat gains or losses and the prevailing space-cooling/-heating systems are not coupled with mechanical ventilation to provide the required air changes.

## 2. Approach

Implementing an effective heating, cooling, and ventilation system is crucial for maintaining indoor air quality (IAQ), thermal comfort, and energy consumption in schools [38]. In this regard, demand control ventilation (DCV) systems offer energy-efficient and sustainable solutions by adjusting ventilation rates based on occupancy and IAQ levels [39] in comparison to traditional mechanical ventilation systems, which supply a constant volume of fresh air regardless of occupancy levels [8].

$CO_2$-based DCV is considered less energy-intensive than other types of DCV systems, because the use of $CO_2$ levels for control is a reliable and accurate indicator of occupancy levels and indoor air quality and leads to energy savings [38]. $CO_2$ sensors can detect the amount of $CO_2$ produced by occupants, which correlates with the occupancy levels, and adjust the ventilation rates accordingly. This means that $CO_2$-based DCV systems can ensure optimal IAQ while minimizing energy consumption and operating costs [40]. The potential for energy savings using a $CO_2$-based DCV strategy is influenced by various factors, including occupant density, variability, and climate [39]. However, as highlighted by Ng et al. [9], it is also important to consider the building code and standard of a particular country, as these provide specific guidelines for minimum ventilation rates required for satisfactory indoor air quality (IAQ).

For the case study under consideration in Malta, where no country-specific guidelines exist for ventilation in school buildings, the authors chose to focus on standard EN16798-1 [41], given that Malta is a Member State of the EU. This standard specifies design values of indoor thermal comfort based on different building categories and occupants' level of expectation.

To ensure optimal indoor thermal comfort, it is necessary to understand the best strategies to ventilate schools in Malta's Mediterranean climate. However, there is a knowledge gap surrounding the most effective approach to achieve this goal while aiming for energy saving.

Unfortunately, due to the baseline being natural ventilation in most schools in Malta, it is inevitable that energy consumption will increase if mechanical ventilation methods are implemented. However, the well-being of the students and their academic performance must be at the forefront of any such analysis. Moreover, the EU Energy Performance of Buildings Directive (EPBD) 2018/844 [42] puts energy efficiency, indoor air quality, and indoor comfort on equal footing.

By means of on-site measurements and building energy modelling, this paper aims to achieve the following:

(1) Assess the IAQ and the thermal comfort of primary school classrooms when using natural ventilation and infiltration only;

(2) Analyse the improvement in IAQ and thermal comfort of primary school classrooms when mechanical exhaust fans are introduced;

(3) Determine whether night purging can be effective in the Maltese Mediterranean climate to precool air temperature before occupation, during summer;

(4) Model and measure whether warmer corridor air can be used to replace stale air in the classrooms by expelling air through DCV-controlled exhaust fans in the classroom while withdrawing corridor air into the classroom to maintain IAQ and reduce energy needed for space heating in winter, i.e., utilising the corridors as a sunspace;

(5) Compare the energy savings and life-cycle costs of continuous running and inverter-driven ventilation fans to determine the most cost-efficient DCV fan mode over the lifetime.

Ultimately, this study seeks to gain a deeper understanding of how best to improve the indoor environment of classrooms by enhancing thermal comfort and IAQ and promoting a healthy learning environment for students in Mediterranean schools based on a real-life case study.

## 3. Case Study Description

This case study was carried out in Malta, which has a hot-summer Mediterranean climate according to the Köppen–Geiger climate classification [43], with very mild rainy winters and dry, warm-to-hot summers. This study was performed on a primary school building located in Siġġiewi, forming part of St. Ignatius College. Table 1 below shows an extract of the long-term monthly averages for the period 1990–2020 [44].

**Table 1.** Selected long-term weather parameters for Malta (1990–2020) [44].

| Month | Av./Max/Min Temperatures (°C) | Av. Bright Sunshine Hours (h) | Wind Speed (m/s) |
|---|---|---|---|
| January | 12.9/15.7/10 | 5.4 | 4.6 |
| February | 12.6/15.7/9.6 | 6.6 | 4.8 |
| March | 14.1/17.4/10.9 | 7.2 | 4.9 |
| April | 16.4/20.1/12.7 | 8.4 | 4.9 |
| May | 20.0/24.3/15.8 | 9.9 | 4.5 |
| June | 24.2/28.8/19.6 | 11.2 | 3.9 |
| July | 26.9/31.7/22.1 | 11.9 | 3.5 |
| August | 27.5/32.0/23.0 | 10.9 | 3.2 |
| September | 24.9/28.6/21.2 | 8.4 | 3.7 |
| October | 21.7/25.0/18.4 | 7.0 | 3.8 |
| November | 17.9/20.8/15.0 | 6.1 | 4.2 |
| December | 14.5/17.1/11.8 | 5.3 | 4.7 |

### 3.1. School Layout and Classroom Selection

The case study school contains several classrooms in all cardinal directions within an angle of ±20°; this enables the study of effective ventilation in all directions. The layout shown in Figure 1 resembles many other school layouts existing in Malta. Two classrooms were chosen from each floor and for each orientation to gauge the effect of wind speed and direction on each façade while taking into consideration solar gains.

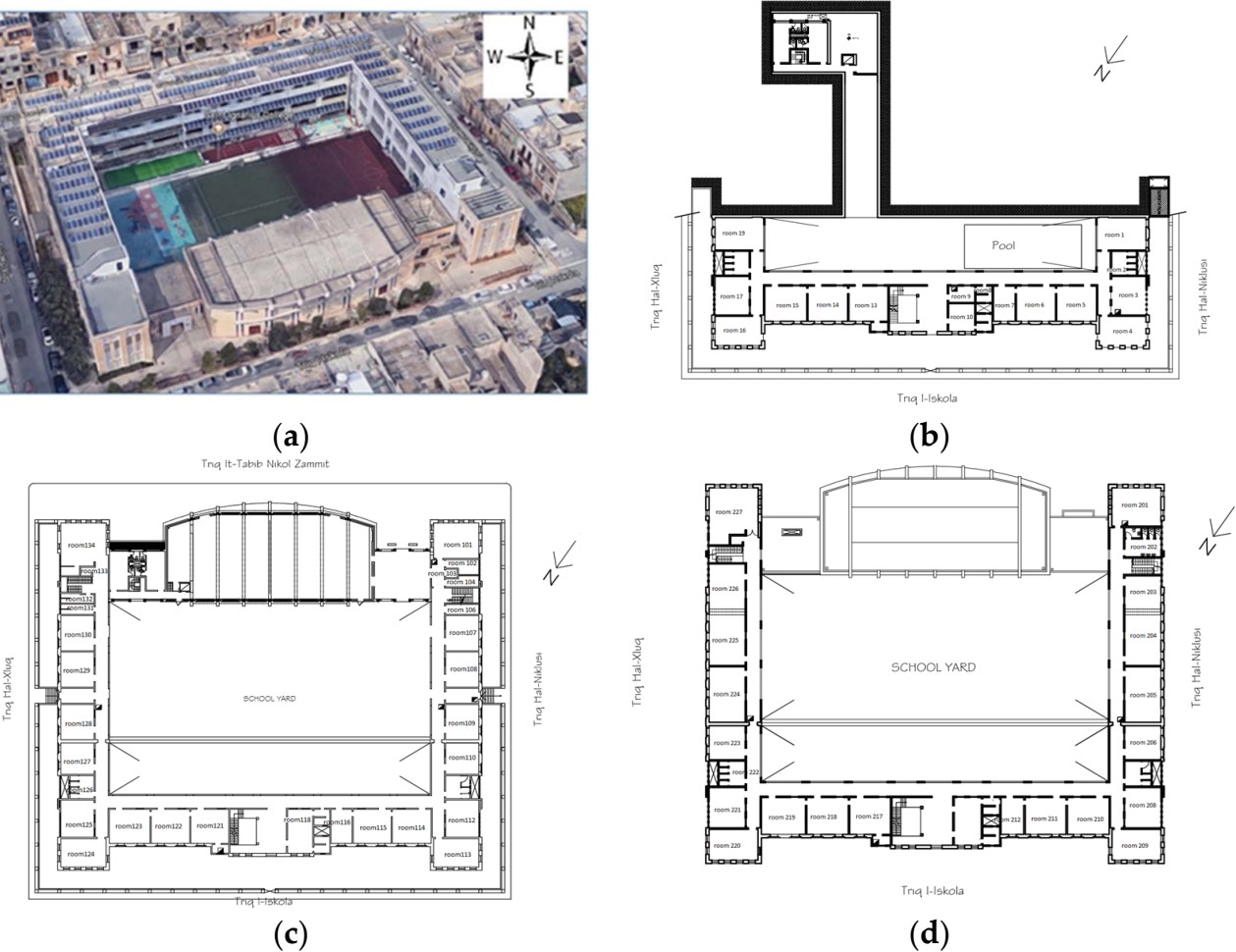

**Figure 1.** (**a**) St. Ignatius College Primary School, Siġġiewi; (**b**) lower-ground-floor plan; (**c**) upper-ground-floor plan; (**d**) first-floor plan.

### 3.2. Ventilation System Upgrades in Retrofitted Classrooms

The school, which previously relied on natural ventilation, was retrofitted with a mechanical exhaust ventilation system to improve the classroom indoor air quality conditions in winter, while continuing to benefit from natural ventilation alone via the opening of windows during the summer months when temperature conditions are more favourable. A building management system (BMS) was installed to control ventilation as well as monitor various parameters using appropriate sensors installed in each classroom such as $CO_2$, ambient and radiant temperatures, relative humidity, occupancy, and air velocity to facilitate this study. A typical classroom is shown in Figure 2a below, while Figure 2b shows the classroom after retrofitting.

Each classroom retrofit consisted of a ventilation system containing a fan, mixing box, controls, and sensors (these include $CO_2$ and thermal sensors), ventilation inlet and exhaust grills, and a control interface, which can be automated or manually bypassed (refer to Figure 2b). Each classroom has external vertical aluminium louvers in front of windows which can be opened and closed either manually or by means of a BMS control system.

The ventilation system (refer to Figure 3) in each classroom includes an exhaust duct fan, an interior circular air valve, and interior mounted grilles. The ventilation fans (TD-1000/200 Silent Ecowatt) are BMS-controlled and vary their air volume discharge between 500 m$^3$/h (or lower) to 1000 m$^3$/h, depending on the required parameters, mainly differential temperature and $CO_2$ levels. The ventilation system gives priority to controlling the $CO_2$ concentration, overriding the other parameters when necessary.

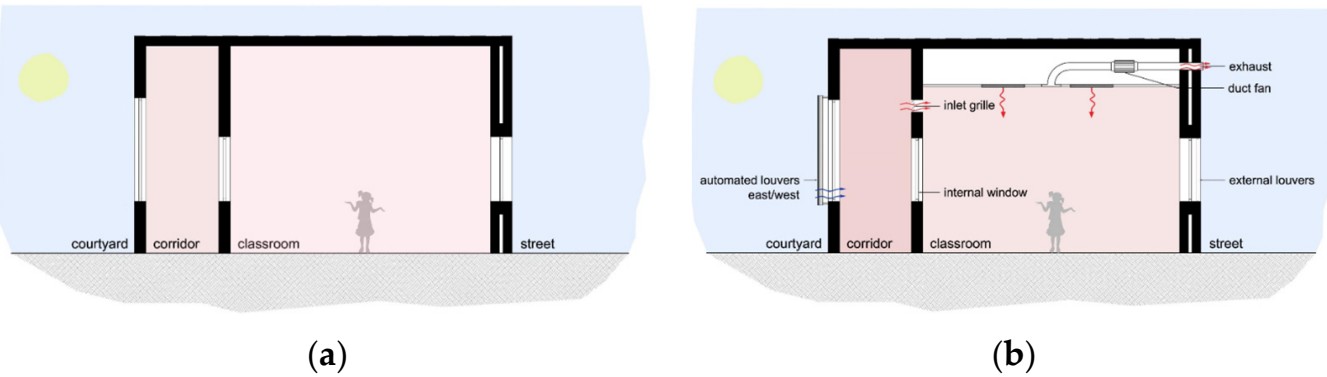

**Figure 2.** (**a**) Typical classroom at St. Ignatius College Siġġiewi Primary School before retrofitting; (**b**) typical classroom at St. Ignatius College Siġġiewi Primary School after retrofitting.

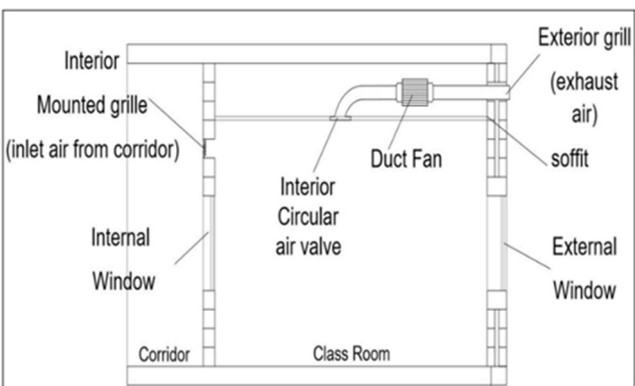

**Figure 3.** Classroom ventilation system.

## 4. Methodology

This study was performed using on-site measurements of temperature and $CO_2$ levels in the classrooms and in nearby corridors and a calibrated building energy model (BEM) for the whole building based on EnergyPlus in DesignBuilder (version 5.0.3.007) software. The following modelling steps were applied:

1. Characterise BEM for the building under study in DesignBuilder in terms of geometry, operation, form, system, and envelope.
2. Calibrate the BEM using hourly on-site measurements of indoor $CO_2$ levels.
3. Model and measure temperature and $CO_2$ levels inside various corridors and classrooms to investigate the following:
   a. The potential of using only natural ventilation and infiltration without mechanical ventilation to achieve the required air changes per hour.
   b. The effectiveness of the mechanical exhaust fan (mechanical ventilation) installed in classrooms to achieve the required air changes per hour.
   c. The night-purging potential using mechanical ventilation for the summer period.
4. The calibrated building energy model was also used for the following:
   a. To assess the potential of sunspace heating for transferring warm air from corridor to classrooms in the winter season using different air inlet configurations.
   b. To assess different modes of mechanical ventilation control in terms of costs.

### 4.1. Measuring Equipment Installed at the School

Sontay GS-$CO_2$-1001 sensors from Sontay, Kent, England were installed in the corridors and classrooms to measure dry bulb temperature (DBT) and $CO_2$ levels. These sensors can

measure DBT with a range of 0 to 40 °C, accuracy: ±0.6 °C, and $CO_2$ level with a range of 0 to 2000 ppm, accuracy: ±50 ppm.

In addition, Sontay TT-1015 sensors from Sontay, Kent, England were installed in the classrooms to measure the "operative" temperature (also known as "comfort" temperature in EN 16798-1), by considering both the air temperature and the surface radiant temperature within a controlled space. This sensor has a temperature range of 0 to 70 °C and an accuracy of ±0.2 °C. The data from these sensors were fed into the building management system (BMS) interface to control the installed ceiling infrared space heaters and ventilation fans.

### 4.2. Calibration and Validation of the Building Energy Model

An EnergyPlus (version 8.5.0.001) BEM was set up in DesignBuilder version 5.0.3.007 as shown in Figure 4. The model's simulated hourly $CO_2$ concentration levels results were validated and calibrated against the monitored actual measured data, in accordance with the ASHRAE guideline [45] for normalised mean biased error (NMBE) and coefficient of variance root mean square error (CVRMSE).

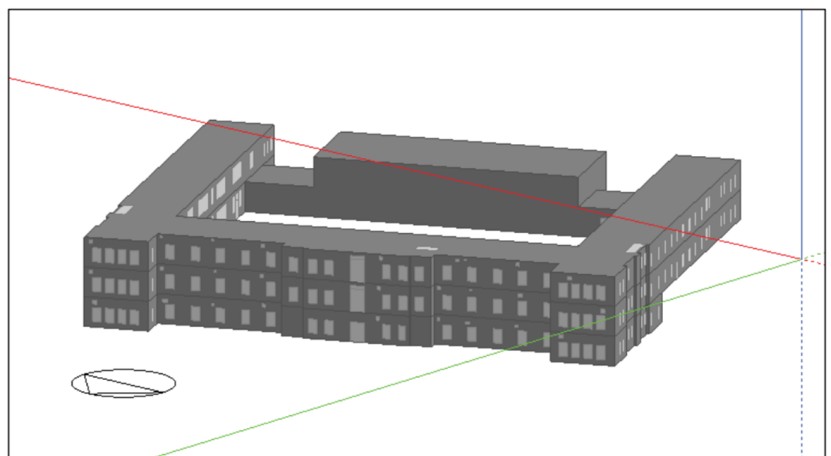

**Figure 4.** DesignBuilder model of St. Ignatius College Siġġiewi Primary School. (The thin green line is the software's inbuilt N-S direction on the ground. The red line indicates the E-W direction on the ground).

For validation, two simulations were conducted, whereby in the first one, the exhaust fan was kept switched off, and in the second one, the exhaust fan was activated whenever the $CO_2$ concentration level reached the set limit of 800 ppm. The following parameters were kept constant for both simulations:

- Infiltration rate at 2 $m^3$/h $m^2$ at 50 Pa.
- Discharge coefficient for open windows and holes: 0.65.
- Typical winter week: 6 to 12 January.
- Occupancy density (persons/$m^2$): 0.5523.
- Activity factor (children): 0.75.

The $CO_2$ concentration simulation was validated by comparing the actual data monitored by the BMS system with the DesignBuilder results for a random classroom (room 210). Table 2 presents the actual and simulated data.

The BMS $CO_2$ concentration data were monitored every 10 min, whilst the simulation data were recorded every hour. The results indicated that the $CO_2$ concentration inside the classroom was at its lowest at 485 ppm during early occupancy hours (08:00) and started increasing thereafter until closure time. During this time, the $CO_2$ concentration in the adjacent classroom corridors was noted to range between 570 to 590 ppm, which was

significantly lower than the adjacent classrooms. The control parameters for validating the results are given below [45]:

$$NMBE = \frac{\sum(CO_2 actual - CO_2 modeled)}{(n-1) * Mean\ (CO_2\ actual)} * 100\% \tag{1}$$

$$CV(RMSE) = \frac{\sqrt{\frac{\sum(CO_2\ actual - CO_2\ modeled)^2}{n-1}}}{Mean\ (CO_2\ actual)} * 100\% \tag{2}$$

where:

- $CO_2$ actual is the parameter measured for each time step (in this case every hour);
- $CO_2$ modelled is the parameter for the modelled value for each time step;
- N is the number of time steps being analysed during the period of evaluation.

The ASHRAE guideline states that a model is considered calibrated if the NMBE is less than 10% and the CV (RMSE) is less than 30% for hourly data. The values obtained for NMBE and CV (RMSE) were 1.28 and 28.56, respectively, indicating that the BEM can be considered as calibrated.

**Table 2.** Comparison of hourly $CO_2$ concentration between actual and simulated results for room 210.

| Time | Monitored Room 210 CO₂ Concentration (BMS Measurement Data) ppm | Simulated Room 210 CO₂ Concentration (DesignBuilder Data) ppm | Simulated Adjacent Corridor CO₂ Concentration (DesignBuilder Data) ppm |
|---|---|---|---|
| 8:00 | 485 | 570 | 570 |
| 9:00 | 790 | 743 | 588 |
| 10:00 | 787 | 805 | 588 |
| 11:00 | 843 | 807 | 588 |
| 12:00 | 790 | 806 | 588 |
| 13:00 | 896 | 906 | 588 |
| 14:00 | 945 | 918 | 589 |
| 15:00 | 872 | 808 | 589 |
| 16:00 | 833 | 675 | 590 |
| 17:00 | 635 | 587 | 575 |
| 18:00 | 575 | 571 | 570 |

The calibrated model was then used to predict the IAQ of the classroom and optimise the whole process for a comfortable learning environment. Figure 5a shows the $CO_2$ concentration when the exhaust fan is off all day, and Figure 5b shows the $CO_2$ concentration when the exhaust fan is switched on and $CO_2$ is kept at 800 ppm by supplying fresher air from the adjacent corridor via a negative pressure differential created between the classroom and the corridor.

*4.3. The Scenarios Investigated Using On-Site Measurements*

This section describes the methodology via on-site measurements used to investigate the following scenarios.

4.3.1. The Potential of Using Just Natural Ventilation or Infiltration for the Required Air Changes per Hour

Using occupancy sensors in classrooms to detect students' presence, $CO_2$ readings were recorded for two representative classrooms, rooms 214 and 227 (position of classrooms marked on Figure 1d). The exhaust duct fan for both classrooms was manually switched off, and the $CO_2$ concentration was recorded. The $CO_2$ concentration was monitored every fifteen minutes for one hour to see if natural ventilation alone was adequate to maintain

the level of $CO_2$ below the 800 ppm set point. Room 214 kept its window partially opened (10%) to test natural ventilation, while room 227 had its windows closed to allow and test ventilation via air infiltration only.

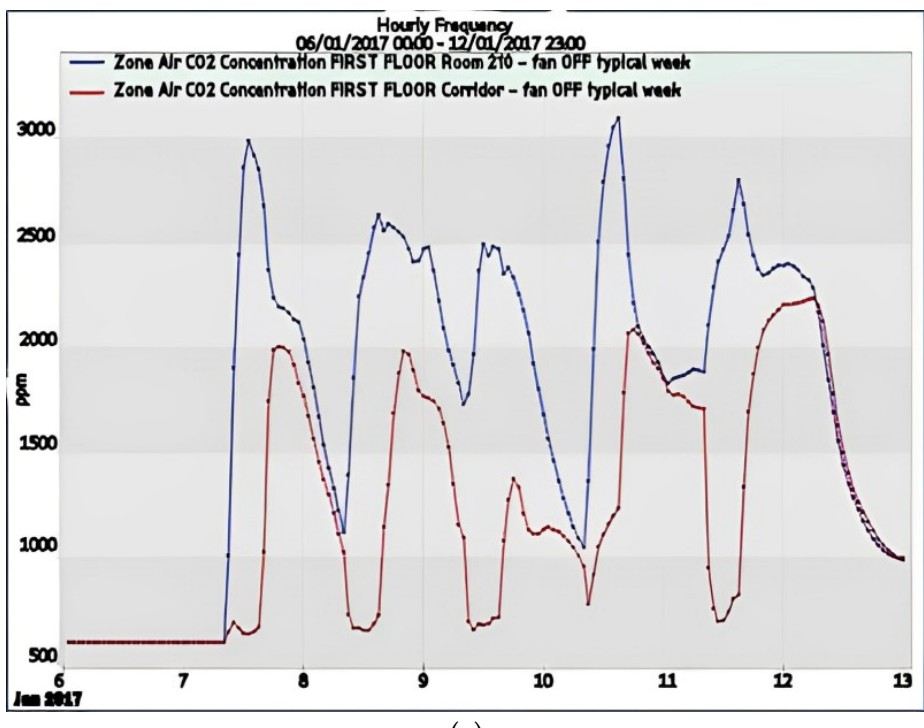

(a)

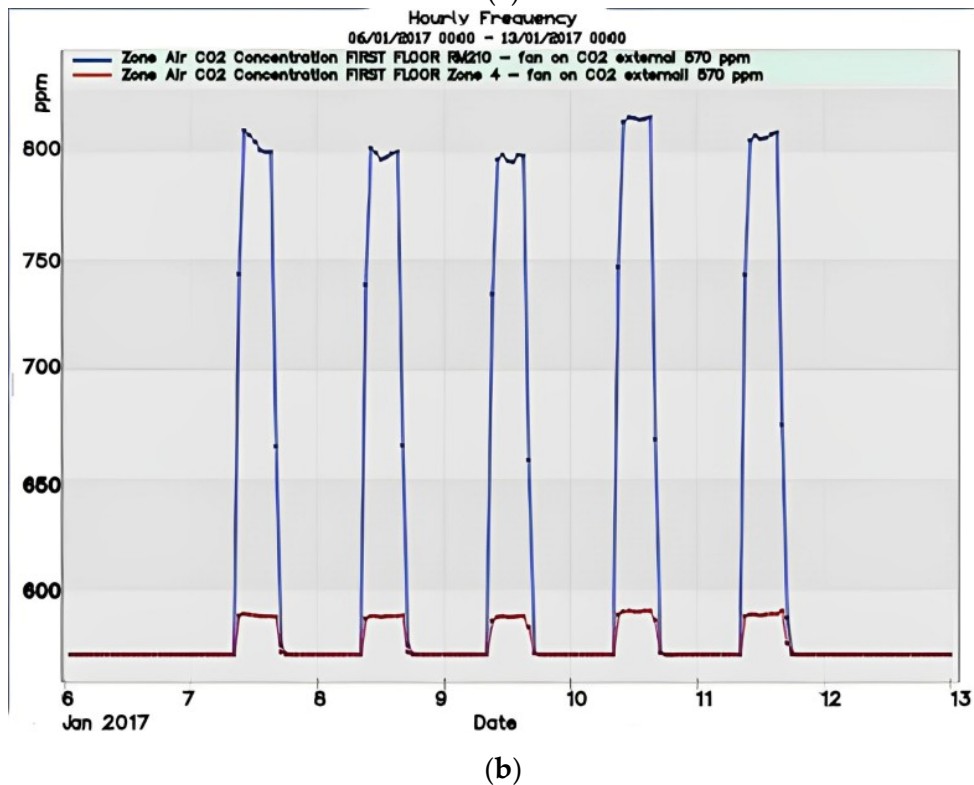

(b)

**Figure 5.** Simulated $CO_2$ concentration in room 210 (blue plot) versus simulated $CO_2$ concentration in adjacent corridor (red plot): (**a**) with exhaust fan off; (**b**) with exhaust fan on.

### 4.3.2. The Effectiveness of the Mechanical Exhaust Fan to Achieve the Required Air Changes per Hour

To determine the effectiveness of the installed exhaust fans in reducing the $CO_2$ concentration levels in classrooms on different floors of the school building, a number of classrooms were chosen, and data were collected from the BMS for one year. This study monitored classrooms on three different levels and in three different cardinal orientations over the course of a year. The exhaust fan $CO_2$ set point was fixed at 800 ppm, and the high limit $CO_2$ set point alarm was set to 1000 ppm.

For the north-facing façade, on the lower ground level, classrooms 5, 6, 13, and 14 were chosen; on the upper ground floor, classrooms 114, 115, 122, and 123 were chosen; and on the first floor, classrooms 210, 211, 218, and 219 were chosen. In the case of the east-facing façade, classrooms 16 and 17 were chosen on the lower ground floor; classrooms 129 and 130 on the upper ground floor; and classrooms 225 and 226 on the first floor. For the west-facing façade, classrooms 3 and 4 were chosen on the lower ground floor; classrooms 107 and 108 on the upper ground floor; and classrooms 203 and 204 on the first floor. Figure 1b shows the specific locations of the monitored classrooms on the lower-ground-floor level, while Figure 1c,d show the locations of the monitored classrooms on the upper-ground level and first-floor level, respectively.

### 4.3.3. Night-Purging Potential Using Mechanical Ventilation for the Summer Period

Several classrooms were selected with different cardinal orientations and at different levels. Each classroom was paired with a benchmark classroom that was not night-purged to ensure that only one parameter was changed in the two adjacent classrooms (Table 3). The goal was to test the effectiveness of night purging in precooling a room during the night in summer before it is occupied again the next day.

**Table 3.** Classrooms under test for the evaluation of night purging.

| Orientation | Level | Night-Purged Classroom | Benchmark Classroom (No Night Purging) |
|---|---|---|---|
| North | Lower ground | 6 | 5 |
| North | Lower ground | 14 | 13 |
| North | Upper ground | 115 | 114 |
| North | Upper ground | 123 | 122 |
| North | First floor | 211 | 210 |
| North | First floor | 219 | 218 |
| East | Lower ground | 17 | 16 |
| East | Upper ground | 130 | 129 |
| East | First floor | 226 | 225 |
| West | Upper ground | 107 | 108 |
| West | First floor | 203 | 204 |

Utilising the exhaust fan present in each classroom, the average classroom temperature during occupied hours was compared to the base scenario without night purging to confirm the hypothesis that night purging is effective. The BMS installed at the school was used to set the purging time from 20.00 to 8.00 the following day, during a typical August week spanning from 20 to 25 August 2018.

Due to security concerns, the external windows were kept closed, and venting was only performed by means of the external exhaust fan vents. This, of course, can reduce the effectiveness of night purging, but if the temperature difference between the purged classroom and the benchmark classroom was appreciable, the trial would be a success and further improvements in the approach could be made. The temperature for each individual room was recorded at 19:45 prior to the starting purging time, and at 08.00 the following day, the exhaust fan was switched off and another set of readings was taken.

*4.4. The Scenarios Investigated Using the Calibrated Building Energy Model*

This section describes the methodology using the calibrated building energy model to investigate the following scenarios.

4.4.1. Use of Sunspace Corridors as Pre-Heated Fresh Air Source to Classrooms during Winter

In Malta, school corridors are neither heated nor cooled. The corridor has a larger glazing-to-wall-area ratio than the classroom. By using the corridor as a sunspace, the heat accumulated there can be transferred to the classrooms, potentially saving on heating costs during the winter season. However, to prevent overheating during the summer, retrofitted shading elements controlled by the building management system (BMS) are implemented in front of corridor windows (Figure 2b). Additionally, as detailed in Section 4.3.3, a nighttime ventilation system was introduced to enhance cooling, contributing to more comfortable temperatures the following morning. To investigate this, a simulation was conducted using the calibrated building energy model to assess the potential of sunspace heating for transporting warm air from corridors to classrooms.

This study specifically focused on classroom 210, which faces the prevailing wind direction during winter. For building energy modelling, the detailed HVAC model option in DesignBuilder was applied. This option specified an exhaust fan placed in the same location as is in the classroom to allow air from the sunspace (corridor) to enter the classroom. The exhaust fan creates a pressure drop inside the classroom, allowing air from the sunspace (corridor) to enter the classroom rather than directly exchanging air with the outside, as is achieved in the simple HVAC mode, Configuration 1. The simulation covered a typical winter week for Malta (20th to 26th January), with results compared between the two configurations.

Four different inlet vent locations, Option 1 to Option 4 as depicted in Figure 6, were also compared in terms of resulting temperature-distribution and air-change effectiveness. CFD simulations were run for each option in DesignBuilder for a typical winter day (9 January) at 09:00 am with the surface boundary conditions imported from the EnergyPlus simulation run performed prior to CFD analysis. This was possible since the EnergyPlus simulations calculate the overall thermal behaviour of the building in terms of inside surface temperatures and natural ventilation flow rates. The default settings and grid provided in DesignBuilder (version 5.0.3.007) enabled a converged solution to be found. Option 1 is the original (actual) inlet vent design simulated for the typical winter week analysis described above, Option 2 is an elongated bottom horizontal vent, Option 3 is an elongated top horizontal vent, and Option 4 is a vertical elongated vent.

4.4.2. Operational Cost Comparison between Different Modes of Mechanical Ventilation Control

The ventilation system in the building was analysed using three different modes of operation for exhaust fan mechanical ventilation control. In Mode 1, the ventilation equipment was integrated with the BMS system and used an on/off $CO_2$ demand ventilation control. Here, the exhaust fan could be set to operate at either on or off positions. All classroom fans were set at 1000 m$^3$/h to comply with the recommended fresh air requirements of 10 L per second per person according to CIBSE Guide A [46], with $CO_2$ level as the priority parameter. When the $CO_2$ level reading from the sensor exceeded 800 ppm, the exhaust fan activated to reduce the $CO_2$ level inside the classrooms. This eliminated the possibility of reaching the set point alarm limit of 1000 ppm. The 800 ppm measured level aligned with EN 16798-1 [41], Category I, catering for spaces occupied by very sensitive and fragile persons. The second control for the fan was based on the classroom temperature in relation to the temperature in the corridor. If the $CO_2$ level inside the classroom was below 800 ppm, the air temperature was below a predefined set point (20 °C), and the temperature in the corridor was higher than that in the classroom by a set temperature difference, then the ventilation fan would operate to draw warmer corridor air into the classroom. As a result, the infrared panel radiative heaters would operate for less time and save energy. Additionally, the ventilation BMS interface had a scheduling function for night purging

during warm summer months based on favourable external temperature conditions from the weather station. If the external temperature was lower than the classroom's temperature within the scheduled time, warmer classroom air was exhausted to bring cooler external air into the school. Mode 2 employed proportional $CO_2$ DCV, functioning similarly to Mode 1 but adjusting fan flow in real time based on varying $CO_2$ levels inside classrooms. The ventilation flow rate increased when $CO_2$ levels were higher inside the classroom and lowered as $CO_2$ levels approached a lower level. Mode 3 is the cheapest and simplest type of control, utilising manual switches or lighting occupancy sensors to switch on the exhaust fan uninterruptedly at a fixed speed during occupancy hours. The external $CO_2$ concentration was also monitored through the BMS weather station and inputted as a value in the simulation parameters to optimise the whole process.

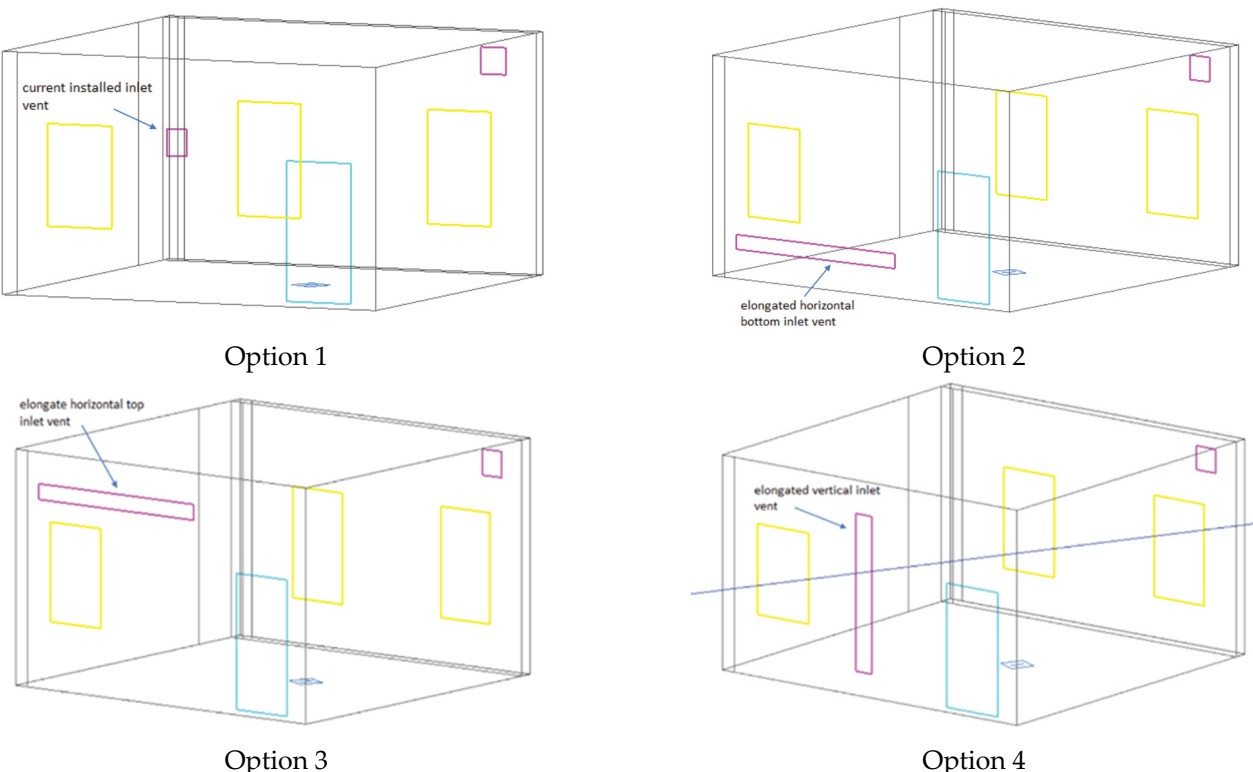

**Figure 6.** Different sizes and positioning of inlet vent (Yellow lines represent windows, blue lines represent solid doors and violet lines represent vents).

A comparison between the three modes of exhaust fan mechanical ventilation control was performed using the calibrated BEM. Classroom no.210 was again chosen as a typical classroom setting for the analysis. The different exhaust fan parameters were input in DesignBuilder using the EnergyPlus simulation engine and the data were analysed. Assumptions were taken regarding the design flow rate, $CO_2$ demand control system, exhaust fan motor efficiency, pressure rise, starting currents, occupancy, and space-cooling and -heating requirements (Table 4).

The simulation focused on the winter period (1 October to 31 March). Modes 1 and 2 required program scripts for on/off and proportional $CO_2$ demand control, respectively, implemented through the Energy Management System (EMS) in EnergyPlus. A model was found to run Mode 3 in the DesignBuilder database, which provides the simplest type of control in which an exhaust fan is uninterruptedly switched "on" at the (design load) fixed speed during occupancy hours via lighting occupancy sensors or manual switches.

**Table 4.** Assumptions.

| Assumptions | Description |
|---|---|
| Design flow rate | Based on CIBSE Guide A [46], the design flow rate for each mode as set at 0.28 m$^3$/s, which can accommodate 28 students with an air change of 10 L/s/person. |
| $CO_2$ demand control system | The $CO_2$ set point was set at 800 ppm, in compliance with standard EN16798-1/2. |
| Exhaust fan motor efficiency | The exhaust fan motor efficiency stayed constant at 67% (full load efficiency) regardless of the variation in flow rate. |
| Pressure rise | The pressure rise stayed constant at 240 Pascals regardless of the variation in flow rate. |
| Operating power | The operating power of the exhaust fan varied directly proportional to the flow rate due to the constant exhaust fan motor efficiency and pressure rise. |
| Surge starting currents | The effects of input surge starting currents on the exhaust motors were negligible. |
| Occupancy | For each simulation and mode of control, 25 persons were assumed to occupy the floor area of 53.9 m$^2$ instead of the classroom design occupancy of 28 persons. This accounted for absentees and provided a more realistic and fair comparison. |
| Space-cooling and -heating requirements | The space-cooling and -heating requirements were assumed to be the same for all ventilation modes. |

## 5. Results and Discussion

### 5.1. Analysis of Using Natural Ventilation or Infiltration

In the analysis of relying solely on natural ventilation or infiltration for the required air changes per hour, the results (see Table 5) demonstrated a notable difference between room 214 and room 227. Room 214, which kept windows partially open, maintained $CO_2$ concentration below 800 ppm, starting at 615 ppm and reaching a maximum of 651 ppm over the testing period. In contrast, room 227, relying solely on infiltration due to closed windows, exceeded the 800 ppm threshold. The initial $CO_2$ level in room 227 was 772 ppm, surpassing 800 ppm within 15 min and reaching 1000 ppm within 45 min. This indicates that infiltration alone is not adequate to keep $CO_2$ concentration below 800 ppm.

**Table 5.** $CO_2$ concentration (measured) in rooms 214 and 227 after one hour.

| Orientation | $CO_2$ Concentration (as Measured) at Specified Time (Outdoor $CO_2$ Concentration at the Time Was Approximately 450 ppm) | | | | |
|---|---|---|---|---|---|
| Time intervals (minutes) | t = 0 | t = 15 | t = 30 | t = 45 | t = 60 |
| Room 214 (ppm): partially opened windows | 615 | 644 | 637 | 651 | 632 |
| Room 227 (ppm): closed windows | 772 | 845 | 930 | 1030 | - |

Drawing from the results of room 214, the findings suggest that natural ventilation via windows can effectively maintain the IAQ in classrooms when outdoor conditions are favourable. However, it is crucial to acknowledge the limitations of relying on natural ventilation for consistently achieving good IAQ. As suggested in the literature [23], natural ventilation through open windows is influenced by human behaviour, making it challeng-

ing to predict and manage. Regular monitoring of $CO_2$ concentration remains essential, with ventilation rates adjusted accordingly. This underscores the importance of having a reliable mechanical ventilation system in classrooms, especially in situations where natural ventilation proves impractical or ineffective.

*5.2. Analysis of Using Mechanical Exhaust Fan*

For analysing the effectiveness of the mechanical exhaust fan to achieve the required air changes per hour, the $CO_2$ concentration levels in the classrooms for each façade when the exhaust fan was used can be found in Table 6. The findings suggest that the performance in classrooms can vary depending on the facing façade, as follows:

- North: 71% of the time below 800 ppm and 6% above 1000 ppm.
- East: 76% of the time below 800 ppm and 7% above 1000 ppm.
- West: 85% of the time below 800 ppm and 3.6% above 1000.

**Table 6.** Hourly $CO_2$ concentration levels in the classrooms for each façade.

| Orientation | Bin | Midpoint | Frequency | Percentage (%) |
|---|---|---|---|---|
| North | 300 | 225 | 0 | 0 |
| | 450 | 375 | 3446 | 13.3 |
| | 600 | 525 | 8911 | 34.3 |
| | 750 | 675 | 6066 | 23.4 |
| | 900 | 825 | 4123 | 15.9 |
| | 1050 | 975 | 1766 | 6.8 |
| | 1200 | 1125 | 797 | 3.1 |
| | 1350 | 1275 | 302 | 1.2 |
| | 1500 | 1425 | 168 | 0.6 |
| | 1650 | 1575 | 135 | 0.5 |
| | 1800 | 1725 | 80 | 0.3 |
| | 1950 | 1875 | 58 | 0.2 |
| | 2100 | 2025 | 122 | 0.5 |
| East | 300 | 225 | 1 | 0 |
| | 450 | 375 | 8378 | 24.1 |
| | 600 | 525 | 11,202 | 32.2 |
| | 750 | 675 | 6774 | 19.5 |
| | 900 | 825 | 3892 | 11.2 |
| | 1050 | 975 | 2204 | 6.3 |
| | 1200 | 1125 | 1278 | 3.7 |
| | 1350 | 1275 | 376 | 1.1 |
| | 1500 | 1425 | 198 | 0.6 |
| | 1650 | 1575 | 130 | 0.4 |
| | 1800 | 1725 | 81 | 0.2 |
| | 1950 | 1875 | 102 | 0.3 |
| | 2100 | 2025 | 192 | 0.6 |
| West | 300 | 225 | 0 | 0 |
| | 450 | 375 | 12,068 | 35.7 |
| | 600 | 525 | 11,161 | 33 |
| | 750 | 675 | 5461 | 16.2 |
| | 900 | 825 | 2690 | 8 |
| | 1050 | 975 | 1169 | 3.5 |
| | 1200 | 1125 | 656 | 1.9 |
| | 1350 | 1275 | 335 | 1 |
| | 1500 | 1425 | 114 | 0.3 |
| | 1650 | 1575 | 59 | 0.2 |
| | 1800 | 1725 | 32 | 0.1 |
| | 1950 | 1875 | 16 | 0 |
| | 2100 | 2025 | 22 | 0.1 |

The results affirm the effectiveness of the installed exhaust fans in reducing $CO_2$ concentration levels and achieving the required air changes per hour. It was also noted that the $CO_2$ concentration is much lower in the corridor than in the classroom. When the exhaust fan operates, high $CO_2$ concentration is expelled to the outside neighbouring street from the classroom, while lower $CO_2$ concentration air is drawn from the internal corridor.

This study underscores the importance of regular monitoring of $CO_2$ levels in classrooms, aligning with the literature recommendations by Schibuola and Tambani [23], and emphasizes the need for adequate ventilation systems for a healthy and comfortable learning environment.

### 5.3. Night-Purging Potential Using Mechanical Ventilation for the Summer Period

When analysing the night-purging potential using mechanical ventilation for the summer period, the temperature differences between the start of purging time at 19:45 and 08.00 the following morning (when the exhaust fan was switched off) were calculated and plotted in Figures 7–9 for both classrooms that were not purged and classroom that were purged. The results showed that the night-purged classrooms had an average temperature between 1 °C and 1.4 °C lower than their counterparts without night purging, regardless of floor level and classroom orientation.

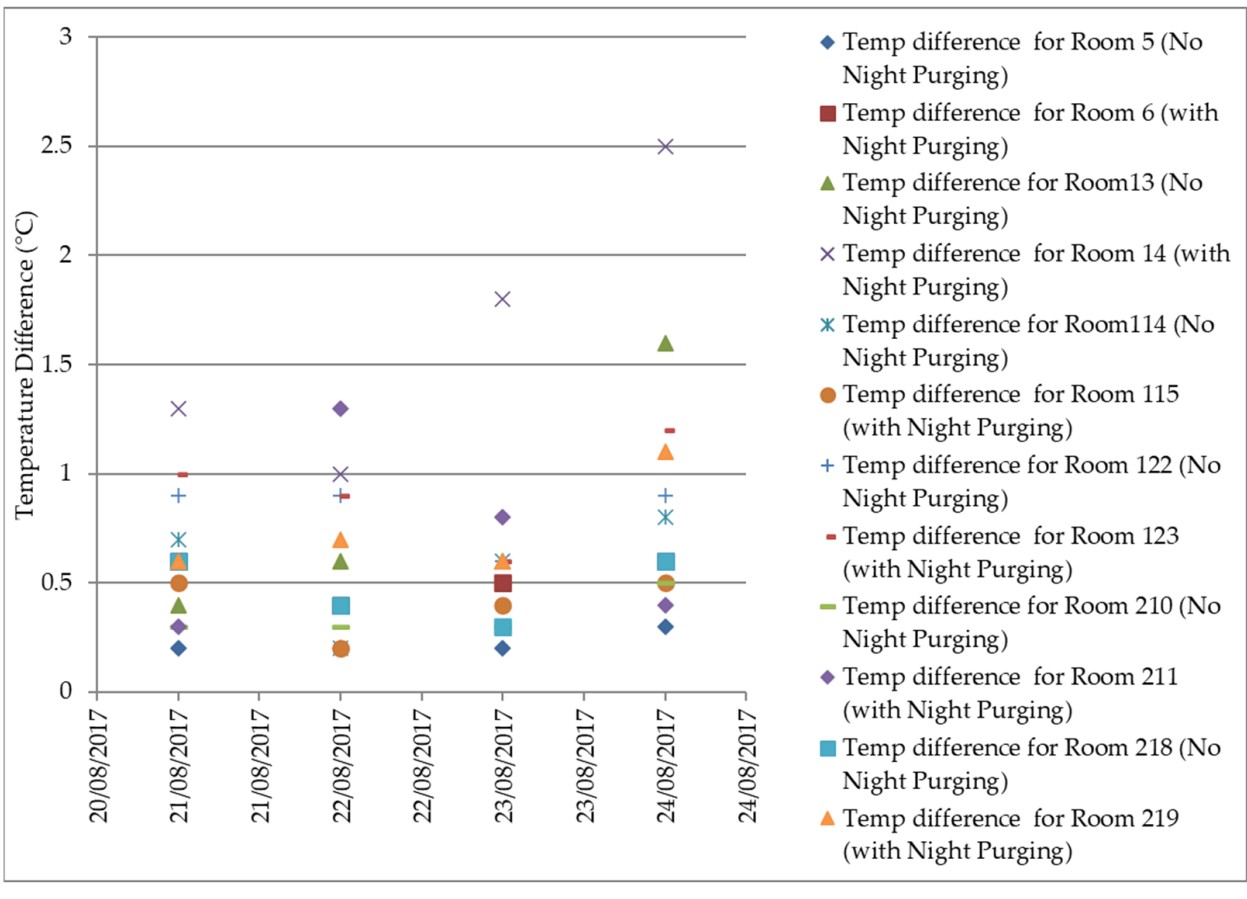

**Figure 7.** Actual (as measured) temperature differences for purged and unpurged classrooms for the north-facing façade.

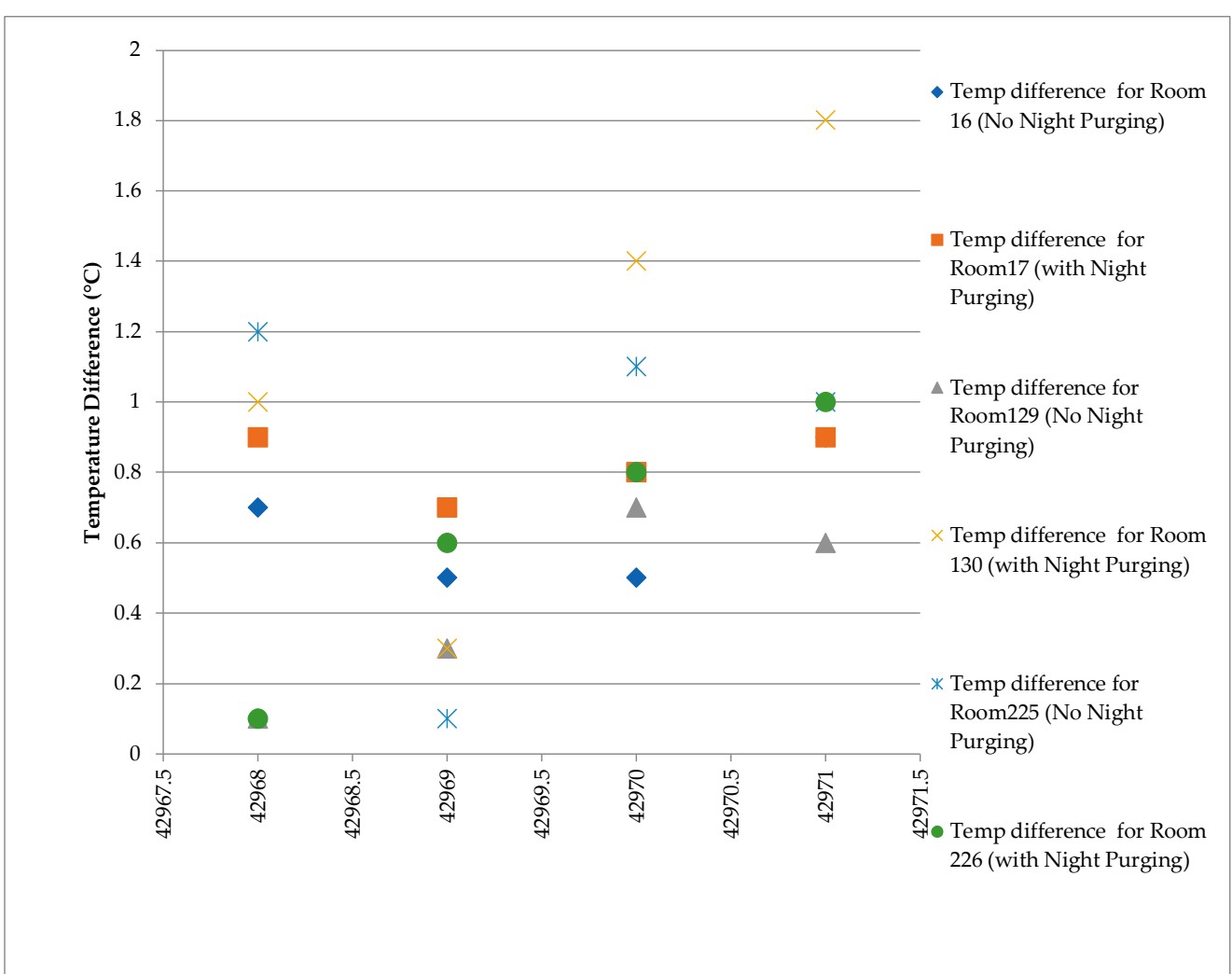

**Figure 8.** Actual (as measured) temperature differences for purged and unpurged classrooms for the east-facing façade.

The findings align with previous studies by Gatt et al. [28] and Arens International [47], showing that night purging is most effective in climates with a large diurnal temperature difference, unlike in the Maltese climate. However, the results also indicate that night purging can be effective even in warmer conditions such as those in Malta, especially in reducing airborne pollutants and facilitating the entry of fresh air.

A notable limitation of the experiment was the closed external windows due to security concerns, potentially diminishing the effectiveness of night purging. Despite this limitation, the results demonstrated a noticeable temperature difference between the night-purged classrooms and the base scenarios. Further research could explore the effectiveness of night purging under different weather conditions and in situations where external windows can be fully opened for ventilation.

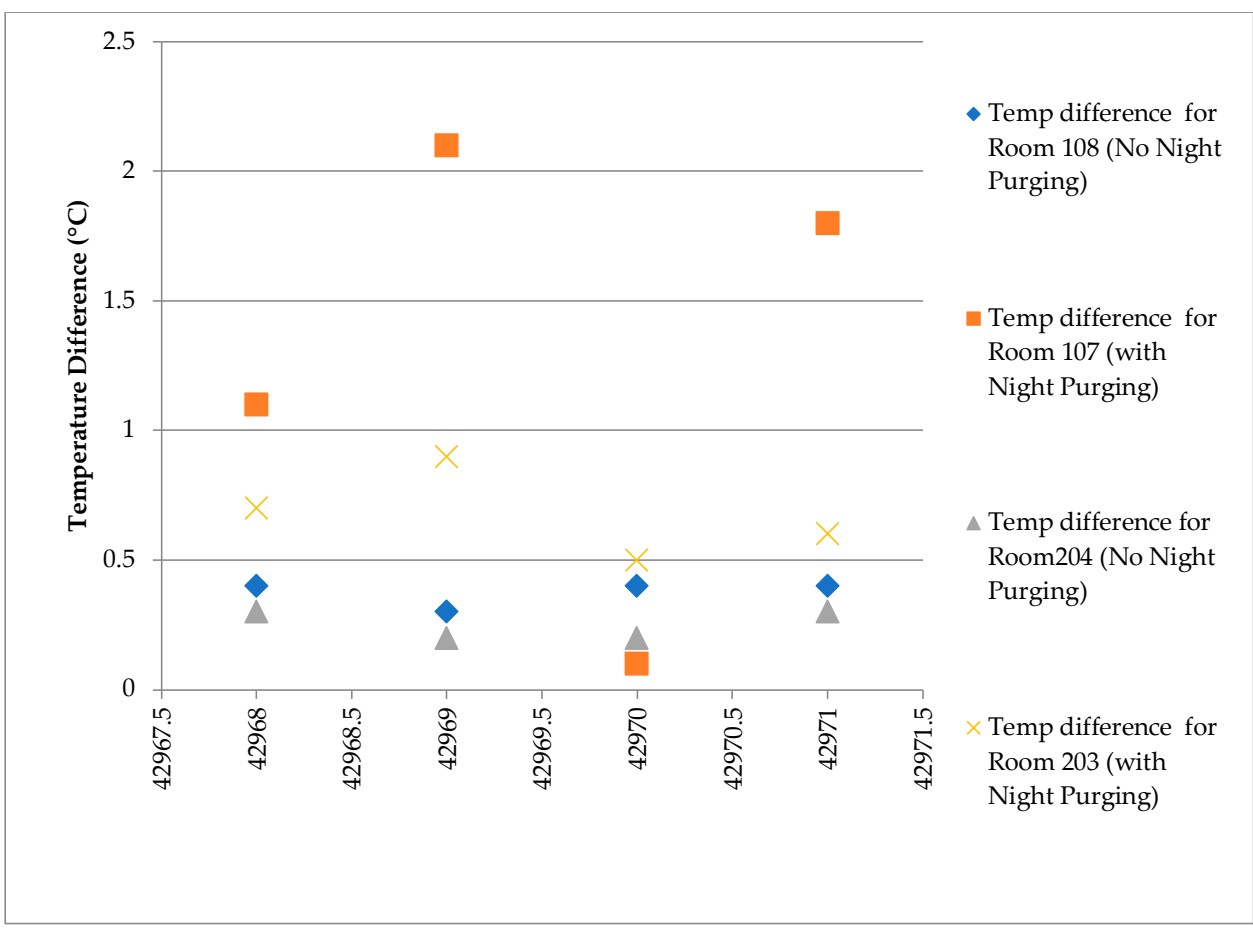

**Figure 9.** Actual (as measured) temperature differences for purged and unpurged classrooms for the west-facing façade.

### 5.4. The Potential of Sunspace Heating for Transporting Warm Air from Corridors to Classrooms in the Winter Season

The potential of sunspace heating for transporting warmer winter air from corridors to classrooms was modelled using the detailed HVAC approach in DesignBuilder that allows the pressure difference between the classrooms and corridors to be modelled. The operative temperatures inside the classrooms were higher than the temperatures in the corridor (Figure 10), due to the classrooms satisfying the required air changes per hour via the adjacent corridor air instead of the colder outside air. Thus, bringing preheated air from the corridor provides an advantage over exchanging classroom air with the outside air. The $CO_2$ level was also found to be below the maximum set point level of 800 ppm, and thus the IAQ requirements could be satisfied.

The CFD analysis for room 210 was first performed using the existing inlet vent configuration (Figure 6—Option 1). CFD slices for operating temperature were analysed for room 210 to check their conformance with thermal comfort criteria. The sunspace-heated air entered the room from the inlet vent and from the opening area of the door and exited through the outside exhaust vent. The resulting operative temperature distribution at 0.6 m above floor level was between 18.3 and 19.9 °C, which was considered satisfactory given the early operating hours and no space heating.

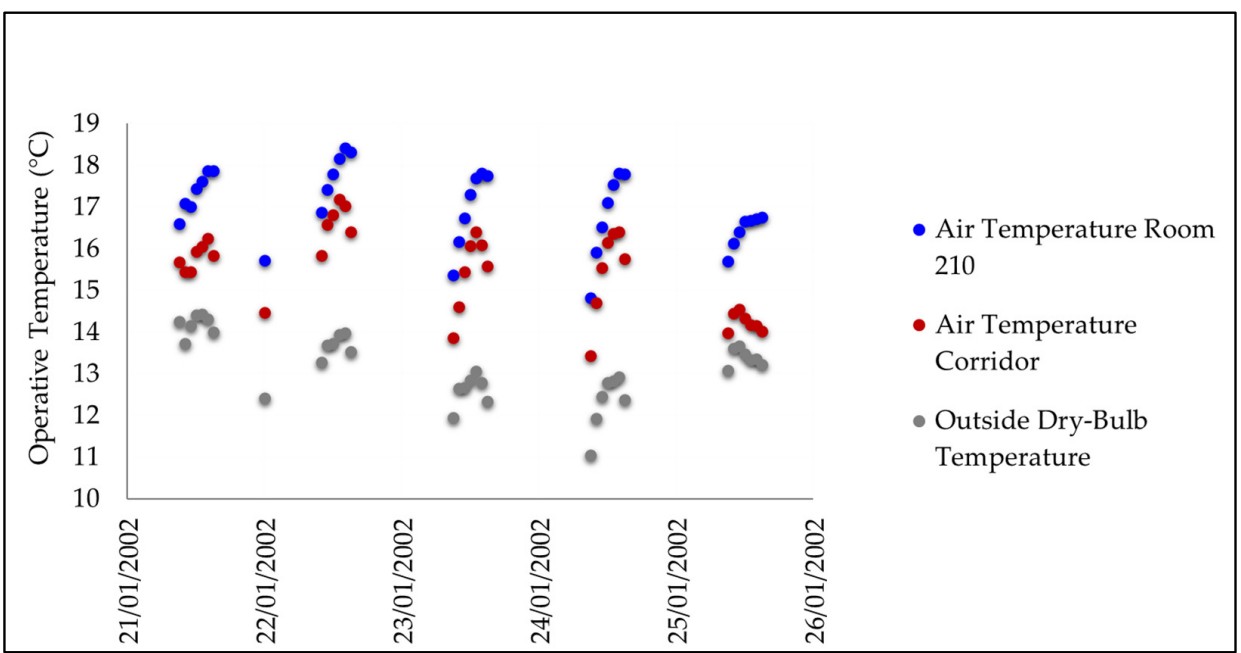

**Figure 10.** Detailed HVAC for classroom 210 (exhaust fan on: simulated) compared to adjacent corridor (Zone 4) with exhaust fan on after $CO_2$ reaches 800 ppm. Ventilation is Mode 1.

Further optimisation of the sunspace concept in terms of inlet location in the divide between classrooms and corridors was modelled using CFD for four options as detailed in Figure 6. Out of all four options, Option 1, the original configuration, gave the best temperature distribution at a height of 0.6 m (Figure 11). The lowest temperature of 18.3 °C was at a more confined and smaller area than the other options.

Although Option 1 with the existing inlet ventilation grid provides the best temperature distribution, other factors, such as age of air, need to be considered to understand which of the four options contributed to the best IAQ distribution. The age of air at any fixed location is the average time that has passed since the air molecules at that location entered the room and is therefore a measure of IAQ. All four inlet vent configurations were simulated, again using DesignBuilder, and it was found that Option 2 with an elongated bottom horizontal vent provided the shortest age of air, ranging between 114 and 342 s. Although this option had a slightly lower average temperature of 18.82 °C at 0.6 m above the floor level compared to Option 1 with an average temperature of 19.38 °C, it had superior IAQ when compared to Option 1, which had an age of air distribution ranging between 198 and 462 s (Figure 12).

Overall, this study showed that the sunspace heating approach could be an effective strategy for transporting warm air from corridors to classrooms during the winter season. However, a detailed HVAC approach is necessary to achieve the desired results. In particular, the inlet location in the divide between the classrooms and corridors needs to be optimised to improve the resulting temperature distribution and IAQ.

The findings of this study have practical implications for the design and operation of school buildings. By utilising the external corridor as a sunspace and bringing preheated air from the corridor, rather than exchanging classroom air with the outside, schools can achieve better thermal comfort and IAQ for their students and teachers without having to rely solely on space heating.

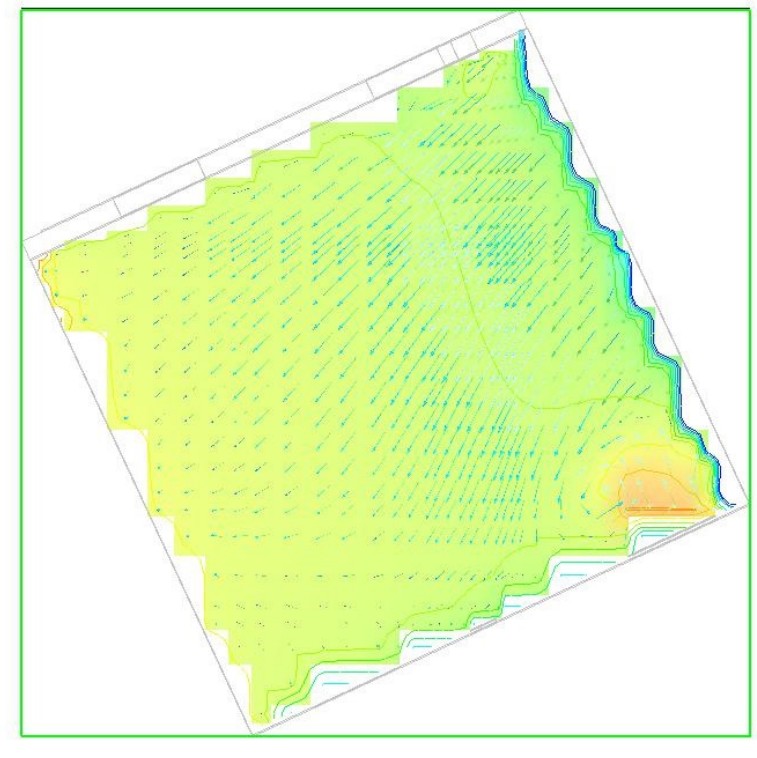

| Velocity | 0.00 | 0.03 | 0.05 | 0.08 | 0.11 | 0.14 | 0.16 | 0.19 | 0.22 | 0.25 | 0.27 | 0.30 | (m/s) |
| Temperature | 17.55 | 17.81 | 18.07 | 18.33 | 18.59 | 18.85 | 19.11 | 19.38 | 19.64 | 19.90 | 20.16 | 20.42 | (C) |

**Figure 11.** CFD simulation (Option 1) at 0.6 m for temperature distribution. The arrows are velocity vectors representing moving air at student desk level of 0.6 m. The colour of the arrows indicates the wind speed in m/s.

### 5.5. Operational Energy Consumption and Financial Feasibility Comparison between Different Modes of Mechanical Ventilation Control

To conduct an operational energy consumption and financial feasibility comparison between the different modes of mechanical ventilation, the $CO_2$ concentration, mass flow rate and electric exhaust fan power were compared for Mode 1, Mode 2, and Mode 3, as explained in Section 4.4.2. For Mode 1 (Figure 13), the zone air $CO_2$ concentration graph shows that the exhaust fan was turned "on" as soon as the concentration exceeded 800 ppm, inducing a mass flow rate of 0.28 $m^3$/s. The fan turned "off" once the $CO_2$ level dropped below this threshold, and the cycle repeated continuously. In Mode 2 (Figure 14), the proportional control operated at the lowest mass flow rate of 0.1 $m^3$/s when the zone $CO_2$ concentration was at its minimum. When the $CO_2$ concentration reached its maximum, the system operated at the maximum flow rate of 0.28 $m^3$/s; otherwise, it operated at a linear proportional control based on the internal $CO_2$ concentration level.

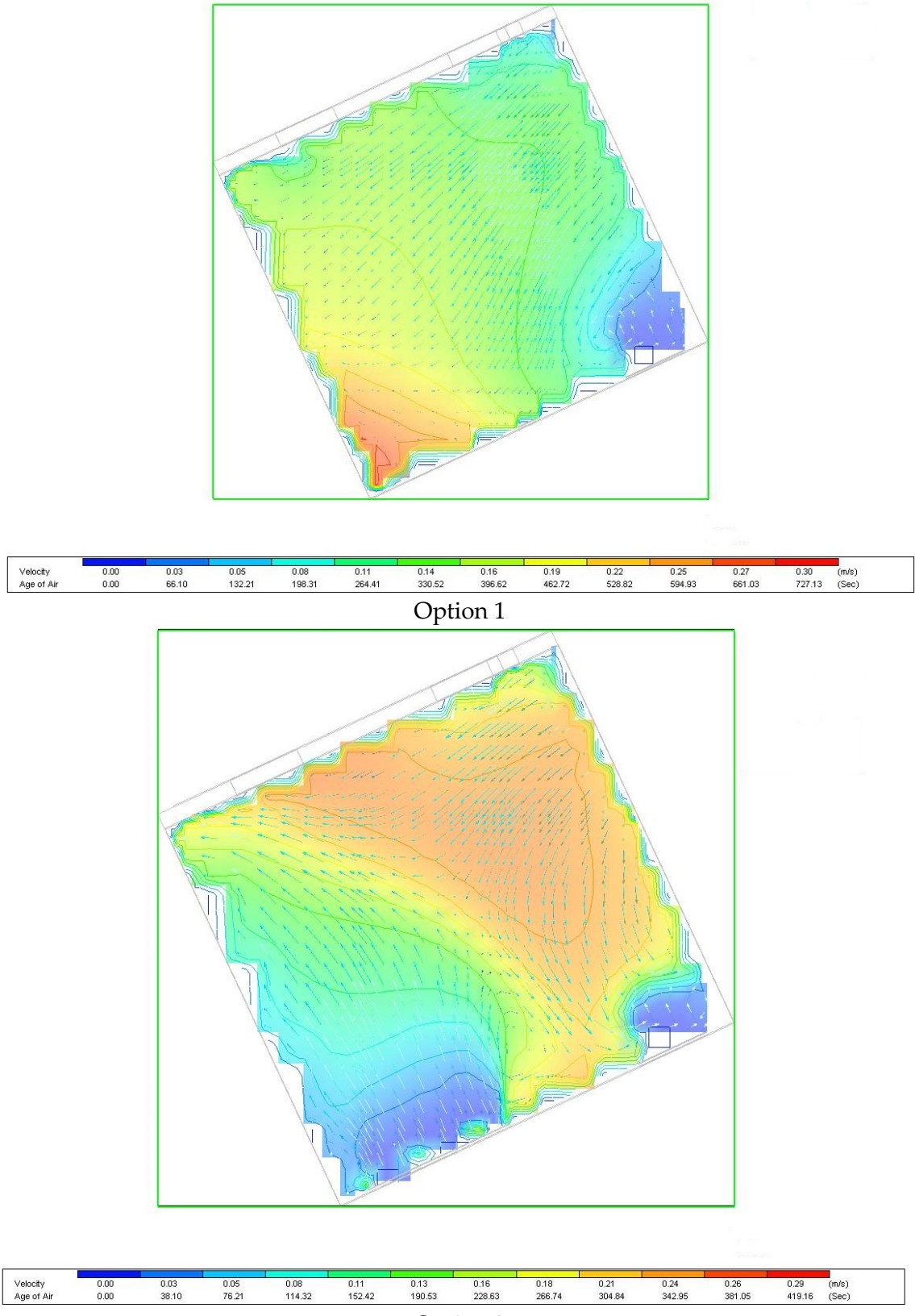

| Velocity | 0.00 | 0.03 | 0.05 | 0.08 | 0.11 | 0.14 | 0.16 | 0.19 | 0.22 | 0.25 | 0.27 | 0.30 | (m/s) |
| Age of Air | 0.00 | 66.10 | 132.21 | 198.31 | 264.41 | 330.52 | 396.62 | 462.72 | 528.82 | 594.93 | 661.03 | 727.13 | (Sec) |

Option 1

| Velocity | 0.00 | 0.03 | 0.05 | 0.08 | 0.11 | 0.13 | 0.16 | 0.18 | 0.21 | 0.24 | 0.26 | 0.29 | (m/s) |
| Age of Air | 0.00 | 38.10 | 76.21 | 114.32 | 152.42 | 190.53 | 228.63 | 266.74 | 304.84 | 342.95 | 381.05 | 419.16 | (Sec) |

Option 2

**Figure 12.** CFD simulation at 0.6 m for age of air distribution. The arrows are velocity vectors representing moving air at student desk level of 0.6 m. The colour of the arrows indicates the wind speed in m/s.

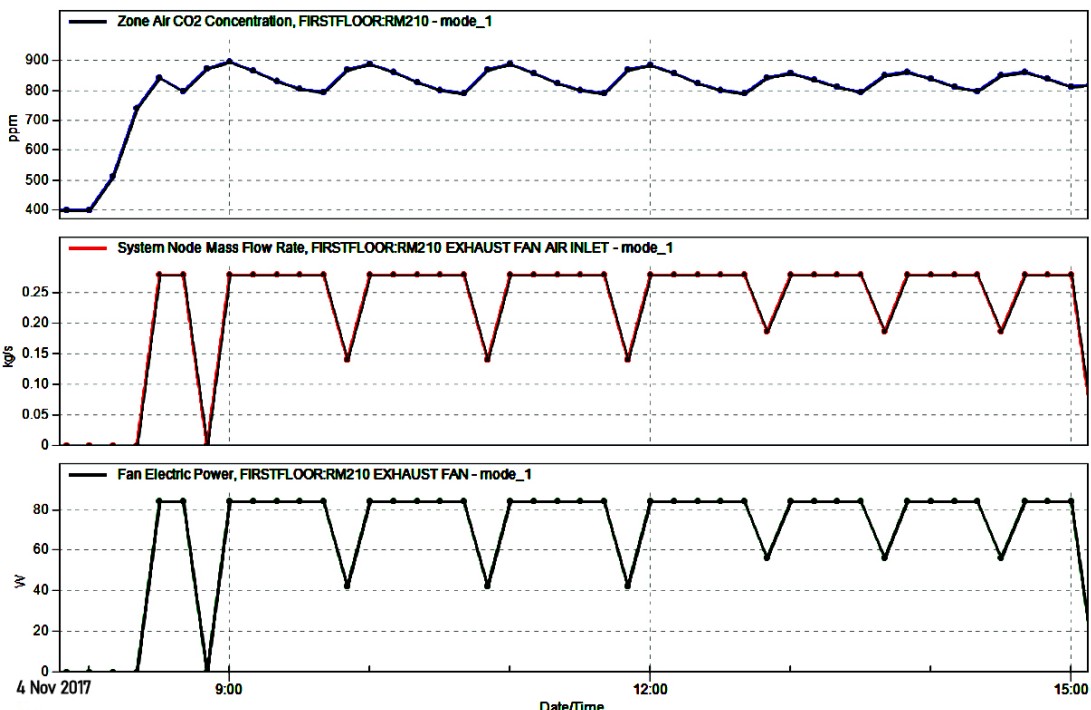

**Figure 13.** Mode 1 zone $CO_2$ concentration, mass flow rate, and fan electric power.

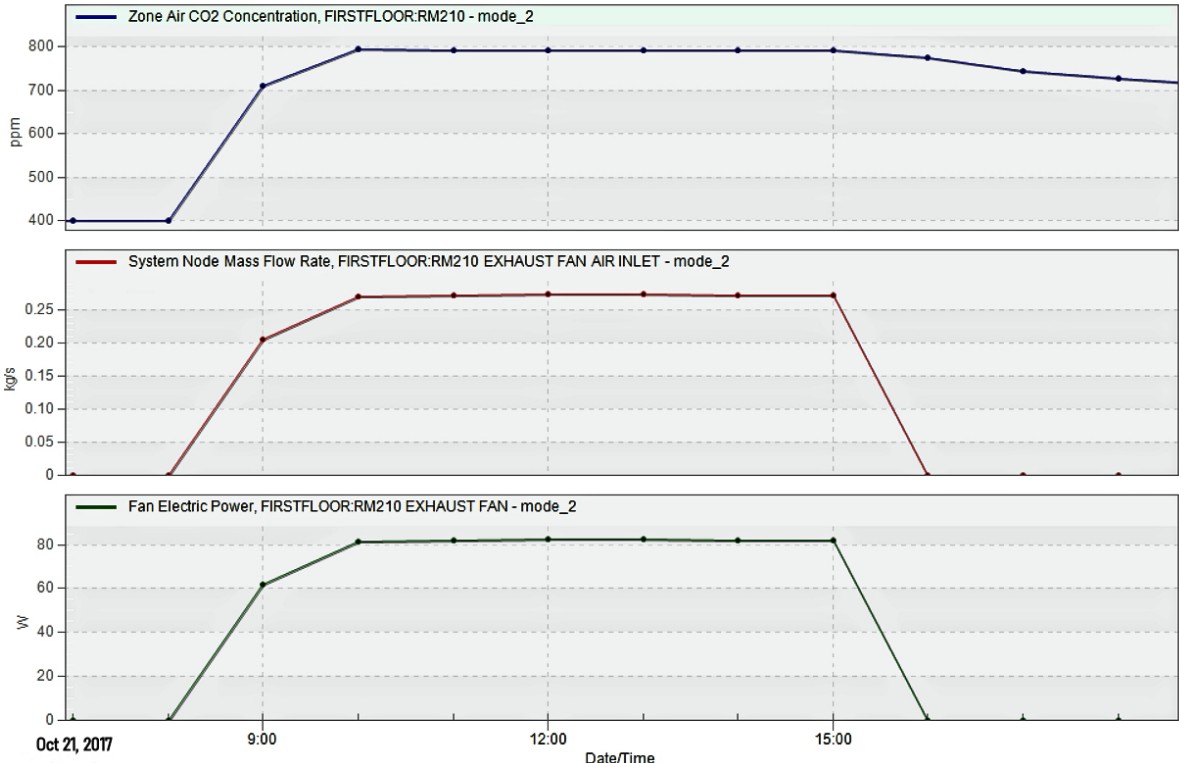

**Figure 14.** Mode 2 zone $CO_2$ concentration, mass flow rate, and fan electric power.

In Mode 3 operation (Figure 15), which operated by switching on at fixed speed during occupancy hours via lighting occupancy sensors or manual switches, the $CO_2$ concentration recorded remained constant at around 700 ppm. On the other hand, in Mode 1, the $CO_2$ concentration varied between 800 ppm and 900 ppm, while in Mode 2, it was kept constant at just below 800 ppm. Although Mode 3 had a slightly higher operating cost, it achieved better IAQ.

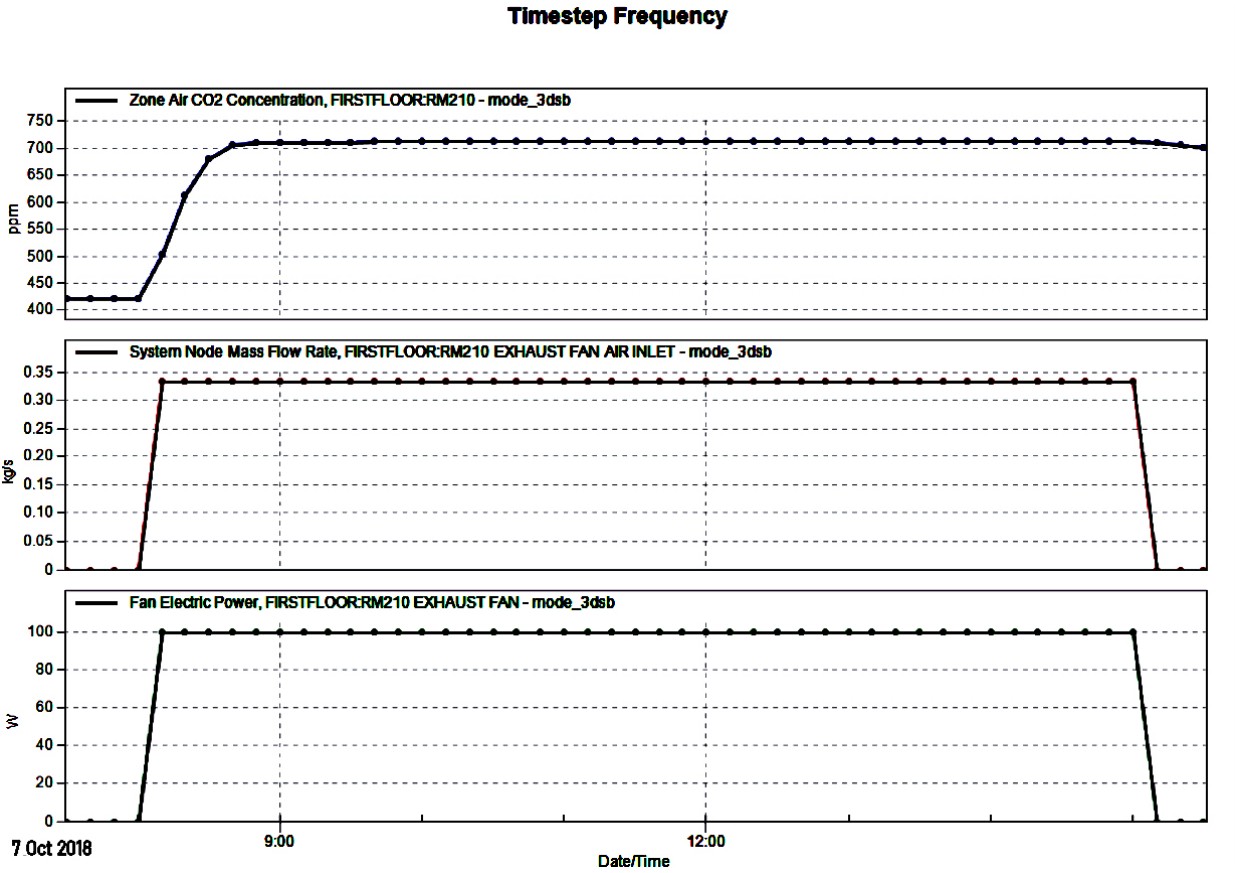

**Figure 15.** Mode 3 zone $CO_2$ concentration, mass flow rate, and fan electric power.

It was observed that when the classrooms are occupied, the fans are switched on most of the time anyway in Mode 1 and Mode 2 to maintain the $CO_2$ concentration within the 800 ppm set point.

Table 7 shows the annual energy consumption and capital, operational, and maintenance costs for the three mechanical ventilation operation modes. The financial global life-cycle costs were calculated using the EN 15459 [48] methodology, covering a 20-year period with a discount rate of 5%. The results indicated that Mode 3 exhibits the most favourable cost effectiveness, as it incurs the lowest global life-cycle costs despite having the highest energy consumption. In essence, this case study demonstrated that the use of variable-speed exhaust fans (Mode 2) did not lead to substantial energy savings and, in fact, resulted in the highest financial global life-cycle cost. It is important to note that these findings are contingent upon occupancy conditions. In this specific case, it was observed that $CO_2$ saturation occurred within the initial 60 min for proportional control. Subsequently, the system continued to operate at design load for approximately 5 h during the occupied period, as depicted in Figure 14, for the various floors.

**Table 7.** Summary results for energy consumption and economic analysis.

| | Energy (kWh/yr) | Capital Cost (EUR)[3] | Operational Cost/yr (EUR)[1] | Maintenance Cost/yr (EUR)[2] | Financial Global Life-Cycle Cost (EUR)[4] |
|---|---|---|---|---|---|
| Mode 1 | 58.6 | EUR 1367 | EUR 8.98 | EUR 27.33 | EUR 2093.20 |
| Mode 2 | 61.8 | EUR 1505.68 | EUR 9.46 | EUR 45.17 | EUR 2598.28 |
| Mode 3 | 77.25 | EUR 1168.88 | EUR 11.82 | EUR 23.37 | EUR 1869.28 |

[1] Operating electrical cost of EUR 0.153 [49] was considered. This cost is calculated from the running mean of the current non-domestic electricity rates for the first 4 bands, which include 5% VAT. [2] Capital costs include VAT of 18%. [3] The maintenance costs for Mode 1 and Mode 3 were taken as 2% whilst for Mode 2, 3% of the capital costs. [4] The disposal costs were assumed to be the same for all modes and therefore not considered for the calculations.

It must also be highlighted that although the on/off control (Mode 1) had a lower energy consumption and a lower global life-cycle cost than proportional control (Mode 2), the impact of starting current was not considered and the IAQ was inferior to Mode 2 control as can be observed by comparing Figure 14 to Figure 15.

## 6. Conclusions

To optimise ventilation in schools for improved comfort and energy efficiency, actual data from the building management system (BMS) were analysed and simulations were conducted using DesignBuilder (5.0.3.007) software with the EnergyPlus simulation engine. Our findings led to the following conclusions:

- During the winter season, occupied classrooms with closed windows did not meet the minimum fresh air changes required for Category I in the European Norm EN 16798-1/2, which is designated for spaces occupied by young children or elderly persons.

- Natural ventilation through the opening of windows in classrooms during the winter season may be sufficient for indoor air quality (IAQ) to conform to the $CO_2$ levels stipulated in EN 16798-1/2 in favourable outdoor conditions but at the cost of higher energy consumption for space heating.

- The installed $CO_2$ demand-driven mechanical exhaust fans at St. Ignatius College Siġġiewi Primary School successfully provided the required air changes for IAQ, as verified by monitoring the indoor $CO_2$ levels from the BMS system in various classrooms on all three floors during occupancy hours for one year.

- By using an exhaust air ventilation system in each classroom and bringing in corridor air via negative pressure differential from inlets spaced in dividing walls between corridors and classrooms, thermal comfort during winter can be improved while reducing the need for space heating. This is because the outside air temperature is on average 3 °C lower than the classroom air temperature during a typical winter week monitored between 21 and 26 January. CFD modelling confirmed that placing the inlets in the dividing wall between classrooms and corridors provided the highest average operative temperature, while an elongated inlet form at the bottom side provided the best age of air distribution, which is a direct measure of IAQ.

- Night purging using mechanical ventilation during the summer period proved somewhat effective. BMS data showed that night-purged classrooms had an average dry bulb temperature from 1 °C to 1.4 °C lower between 19:45 and 08:00 during a typical summer week monitored from the 20 to the 25 August when compared to the benchmark classrooms.

- Mode 3, the cheapest and simplest type of exhaust fan control, consisting of an exhaust fan uninterruptedly switched "on" at maximum fixed speed during occupancy hours, showed better financial feasibility when compared to DCV on/off (Mode 1) and DCV proportional (Mode 2) control. Further studies are necessary to consider other aspects when comparing these different modes, including starting load current, systems lifetime, different fan speeds, and an acoustic comfort comparison.

**Author Contributions:** Conceptualization, D.G.; methodology, D.G.; formal analysis, M.C.; investigation, M.C.; resources, K.R. and C.Y.; data curation, M.C., K.R. and D.G.; writing—original draft preparation, M.C. and K.R.; writing—review and editing, K.R., D.G. and C.Y.; supervision, C.Y. All authors have read and agreed to the published version of the manuscript.

**Funding:** The refurbishment at the Siġġiewi primary school was partly financed by the European Regional Development Fund (ERDF 2007-2013), project reference ERDF342. entitled "Renovating Public Buildings to Increase Energy Efficiency and Reduce GHG (Phase 1)". However, this research received no funding.

**Institutional Review Board Statement:** Not applicable.

**Informed Consent Statement:** Not applicable.

**Data Availability Statement:** The data presented in this study are available on request from the corresponding author. The data are not publicly available due to technical limitations of the BMS.

**Acknowledgments:** We would like to acknowledge the support of the Energy and Water Agency for providing access to the BMS and data bank, as well as the Ministry for Education and the Administration of the St. Ignatius College Primary School Siġġiewi for providing access to the premises and logistic support.

**Conflicts of Interest:** The authors declare no conflict of interest.

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
