# Peer review of "Optimising Mechanical Ventilation for Indoor Air Quality and Thermal Comfort in a Mediterranean School Building"

_sustainability, doi:10.3390/su16020766_

Round 1
Reviewer 1 Report
Comments and Suggestions for Authors
The structure of the manuscript is clear, however I think it lacks originality and, above all, incisiveness. I suggest major revisions to go deeper with data elaboration, with particular attention to the energy aspect (as mentioned in the title), both in terms of energy demand as well as of energy saving potential. Here are some comments I hope will help authors to improve the quality of the manuscript:
- as regards the three modes of operation for exhaust fan mechanical ventilation control (paragraph 4.4.2 and 5.4), it is written that the CO2 concentration threshold was set at 800 ppm. Is there a reason why there was no hysteresis programmed (at least, it seems from the manuscript)? For instance, setting a threshold at 800 ppm for the switching on of the fan and a lower threshold (i.e., 700-750 ppm) for the switching off. Would it have changed anything?
- from the title I would have expected much deeper consideration of the energy aspects. The whole manuscript is focused on indoor air quality, whereas the energy performance and the possibility of reducing the energy demand is vaguely mentioned at the end of the manuscript when comparing the operating times of the three different modes.
- potential of sunspace heating of corridors. It seems you have not considered the negative effect it might have during summer. In cases like this I think it is of utmost importance, since it is the design of the building that affects the behavior, positively during winter but most probably negatively during summer.
- In general, paragraph 5.4 is too superficial. Both the energy as well as financial aspects are treated too generically and could/should have been addressed in much more depth to give completeness to proposed work.
As minor comments:
- some figures have low quality (i.e., fig 2, fig 5). Replace the figures with images at higher definition
- figure 7: there are more dots in the graph than in the legend, I think some descriptions are missing
- figure 10: instead of describing colors in the figure caption, a legend would be more useful and easier to read
- figure 11 and 12: legend is difficult to read
Reviewer 2 Report
Comments and Suggestions for Authors
The main drawback of this work is the way it is presented, there is no clear structure in the writing. It seems to be written as a technical report and not as a scientific article. In addition, the paper is very long, which can make the reader lose interest in it.
- The results are not clearly presented and the images are of poor quality.
- It does not mention the CDF conditions it uses.
-Much of the information could be synthesised with block diagrams.
If the authors could improve the presentation of their work it could be considered for publication in sustainability.
Reviewer 3 Report
Comments and Suggestions for Authors
The analysis was prepared and carried out in great detail.
The writing of units requires correction, because in some of the values the unit is written immediately after the number, and in the remaining values a space is inserted between the number and the unit.
Reviewer 4 Report
Comments and Suggestions for Authors
Dear Authors,
Your study is interesting and contains a certain novelty. The study results are presented and described in sufficient detail. Generally, the manuscript can be approved for publication if some minor comments are eliminated.
1) I would recommend presenting the values of the climate parameters in winter and summer in Malta, as well as the sufficient indoor air conditions in the form of a table. In the article, these parameters are presented in the text form description without exact values when describing the climate in Malta and in the form of a link to EU regulatory documents when describing indoor air conditions.
2) It is worth showing the deviation of the measured values of the parameters in Table 1 (p.9) from their theoretical ones, i.e., to present the measurement error.
3) Figs.1 and 5 should be made with a higher resolution (distinct). The text in Figs.11 and 12 is too small and difficult to read.
4) The numbering of sections is broken. Paragraph 5.4. is presented twice on p.18 and p.22.
Round 2
Reviewer 1 Report
Comments and Suggestions for Authors
I would like to thank the authors for having addressed each of my comment and modified the manuscript accordingly. I do not have anything else to suggest, except for some images that are of low quality, but I presume this will be fixed with the publication.
Reviewer 2 Report
Comments and Suggestions for Authors
The authors have responded to the respective comments